# Assessment of the RANTES Level Correlation and Selected Inflammatory and Pro-Angiogenic Molecules Evaluation of Their Influence on CRC Clinical Features: A Preliminary Observational Study

**DOI:** 10.3390/medicina58020203

**Published:** 2022-01-28

**Authors:** Sylwia Mielcarska, Agnieszka Kula, Miriam Dawidowicz, Paweł Kiczmer, Magdalena Chrabańska, Magdalena Rynkiewicz, Daria Wziątek-Kuczmik, Elżbieta Świętochowska, Dariusz Waniczek

**Affiliations:** 1Department of Medical and Molecular Biology, Faculty of Medical Sciences in Zabrze, Medical University of Silesia, 19 Jordana, 41-800 Zabrze, Poland; eswietochowska@sum.edu.pl; 2Department of Oncological Surgery, Faculty of Medical University of Silesia, 41-808 Katowice, Poland; d201070@365.sum.edu.pl (A.K.); d201069@365.sum.edu.pl (M.D.); dwaniczek@sum.edu.pl (D.W.); 3Department and Chair of Pathomorphology, Faculty of Medical Sciences in Zabrze, Medical University of Silesia, 13-15 3 Maja, 41-800 Zabrze, Poland; pkiczmer@sum.edu.pl (P.K.); hannaonyszczuk@sum.edu.pl (M.C.); mrynkiewicz@sum.edu.pl (M.R.); 4Department of Cranio-Maxillo-Facial Surgery, Faculty of Medical Sciences in Zabrze, Medical University of Silesia, 20-24 Francuska, 40-027 Katowice, Poland; dkuczmik@sum.edu.pl

**Keywords:** RANTES, PD-L1, colorectal cancer

## Abstract

*Background and Objectives*: Assessment of RANTES level and concentrations of inflammatory cytokines: programmed death ligand 1 (PD-L1), interferon gamma IFN-γ, tumor necrosis factor alpha (TNF-α), transforming growht factor β (TGF-β) (and angiogenesis factors: vascular endothelial growth factor A (VEGF-A) and vascular endothelial growth factor C (VEGF C) in tumor and margin tissues of colorectal cancer (CRC,) and evaluation of RANTES influence on histopathological parameters (microvessel density (MVD), budding, tumor-infiltrating lymphocytes (TILs)), in relation to patients’ clinical features. *Materials and Methods*: The study used 49 samples of tumor and margin tissues derived from CRC patients. To determinate the concentration of RANTES, PD-L1, IFN-γ, TNF-α, TGF-β, VEGF-A, and VEGF-C, we used the commercially available enzyme-linked immunosorbent assay kit. Additionally, RANTES and PD-L1 expression was assessed with the use of IHC staining in both tumor cells and TILS in randomly selected cases. MVD was assessed on CD34-stained specimens. The MVD and budding were assessed using a light microscope. *Results*: We found significantly higher levels of RANTES, PD-L1, IFN-γ, TNF-α, TGF-β, VEGF-A, and VEGF-C in the tumor in comparison with the margin. The RANTES tumor levels correlated significantly with those of PD-L1, TNF-α, TGF-β, VEGF-A, and VEGF-C. The RANTES margin levels were significantly associated with the margin levels of all proteins investigated—PD-L1, IFN-γ, TNF-α, TGF-β, VEGF-A, and VEGF-C. Additionally, we observed RANTES- and PD-L1-positive immunostaining in TILs. In a group of 24 specimens, 6 different CRC tumors were positive for RANTES and PD-L1 immunostaining. The IFN-gamma concentration in both tumor and margin and TGF-β in tumor correlated with TILs. TILs were negatively associated with the patients’ disease stage and N parameter. *Conclusions*: RANTES activity might be associated with angiogenesis, lymphogenesis, and immune escape in CRC. RANTES is an important chemokine that is a part of the chemokine–cytokine network involved in the modulation of TME composition in CRC. Further research may verify which processes are responsible for the associations observed in the study.

## 1. Introduction

Colorectal cancer (CRC) is one of the most prevalent cancers worldwide. It is the second and third most common cancer in females and males, respectively, and is accompanied by a high rate of morbidity and mortality. Over 70% of CRC cases are sporadic, 20% of cases have an associated hereditary component, and less than 5% of cases are inherited (Lynch Syndrome, 2–5%) [1]. Recently, the knowledge of the epidemiology, etiology, molecular biology, and clinical aspects of CRC has improved considerably. Nevertheless, 1.8 million new cases are diagnosed annually worldwide. CRC is often diagnosed at advanced clinical stages, and about 900,000 individuals die from this malignancy [2]. A 60% increase in CRC cases is expected worldwide by 2030. The standard treatment for CRC patients is surgery, radiation, chemotherapy, or a combination of these therapies. Additionally, immunotherapy is becoming an attractive option compared with conventional chemotherapy for CRC. Many of the dependencies in CRC are still misunderstood. A better understanding of the mechanisms of this malignancy is essential for the development of modern, effective therapies. The search for new molecules involved in CRC development seems to be crucial [3].

Many cancers represent a paradigm for the link between inflammation and oncogenesis, including CRC. Inflammation is associated with the accumulation of various immune cells and inflammatory mediators, such as cytokines, chemokines, and growth factors. Several studies suggest that chronic inflammation promotes tumor development and leads to the inclusion of inflammation in a characteristic component of tumorigenesis.

Until recently it was believed that the inflammatory chemokines constitute only an antitumor barrier, and these were viewed mainly as indispensable points of regular of immunity and inflammation. Updated information indicates that chemokines may play a very important role in tumor progression, being components of their microenvironment [4]. Tumor cells via chemokine secretion adapt T cells, monocytes, myeloid cells, fibroblasts, and mesenchymal stromal cells (MSCs) from adipose tissue, and stimulate them by different mechanisms to become immunosuppressive T regulatory cells (Tregs), tumor-associated macrophages (TAMs), myeloid-derived suppressor cells (MDSCs), cancer-associated fibroblasts (CAFs), and cancer-associated adipocytes [5]. Tumor cells can take control of chemokine networks to support tumor progression. This phenomenon can be observed in the context of the CCL5/CCR5 axis. 

CCL5 (RANTES) belongs to the C-C chemokine family and plays an active role in recruiting a variety of leukocytes into inflammatory sites, including T cells, macrophages, eosinophils, and basophils. CCL5 is a target gene of NF-κB activity and is expressed by T lymphocytes, macrophages, platelets, synovial fibroblasts, tubular epithelium, and certain types of tumor cells. The RANTES activity is mediated through its binding to CCR1, CCR3, and mainly CCR5. 

The major functions of this chemokine in tumor development are poorly understood. CCL5 production is relevant to inducing proper immune responses against tumors, but, on the other hand, CCL5 is associated with cancer progression and metastasis [6]. CCL5 by interaction with CCR5 can support tumor progression via pleiotropic effects, including by acting as growth factors, stimulating angiogenesis, enhancing tumor cell migration (metastasis formation, modulating the extracellular matrix, inducing the recruitment of additional stromal and inflammatory cells, decreasing the cytotoxicity of DNA-damaging agents, and taking part in immune evasion mechanisms via inducing the immunosuppressive polarization of macrophages [5].

It has been reported that CCL5 is overexpressed in colorectal cancer and plays a crucial role in immune escape of tumor cells. Shengbo Zhang et al. examined that CCL5-deficiency could upregulate PD-1 and PD-L1 expression and reduce the resistance to anti-PD-1 antibody therapy in the CRC mouse model [7]. That study also proved that knockdown of RANTES was associated with the reduction of tumor growth, metastasis and apoptosis of tumor-infiltrating CD8+ T cells. The proangiogenic activity of RANTES can be supported by increasing migration of endothelial cells, spreading, neovessel formation, and secretion of vascular endothelial growth factor (VEGF). 

Many factors are involved in the regulation of CCL5 expression. Researchers have pointed out that there is a relationship between CCL5 levels and inflammatory molecules. TNF-α and IFN-γ have the potential to enhance CCL5 expression by TLR3 signaling. In this mechanism, TNF-α may activate NF-κB, in cooperation with TLR3 signaling. IFN-γ may stabilize CCL5 mRNA up-regulated by TLR3 [8]. IFN-γ and TNF-α are among the most important pro-inflammatory cytokines involved in the recruitment of immune cells to the TME [9,10]. The TILs (tumor-infiltrating lymphocytes) parameter is associated with immune status and its increase is a positive prognostic factor in CRC [11].

Transforming growth factor-β1 (TGF-β1) is a pleiotropic cytokine that also regulates CCL5 levels. TGF-β1 inhibits RANTES expression mediated by β-catenin-triggered blockade of NF-κB signaling [12]. Epithelial to mesenchymal transition (EMT) of tumor cells is reflected by budding and is considered an additional prognostic factor in CRC [13]. Tumor budding is described as the presence of a single cancer cell or clusters consisting of four cells or less at the tumor invasive front [14]. 

In this study, we wanted to investigate the correlations between RANTES and selected factors in the context of immune response and angiogenesis processes in colorectal cancer. Additionally, our aim was to investigate expression of RANTES and PD-L1 in selected cases with the use of immunostaining to determine which cells in the tumor environment produce these proteins. We also assessed whether the levels of RANTES, TNFα, IFNγ, TGF- β1, PD-L1, VEGF-A, and VEGF-C were associated with some histopathological parameters: MVD (microvessel density), budding, and tumor-infiltrating lymphocytes. MVD has been reported to be an independent prognostic factor in many cancers, but in CRC the results are inconclusive [8].

The levels of RANTES, TNFα, IFNγ, TGF- β1, PD-L1, VEGF-A, and VEGF-C were measured using ELISA tests; for 24 randomly selected cases RANTES and PD-L1 immunostaining was performed. These data and other histopathological parameters (MVD, TILs, budding) were correlated with the patients’ clinical features. 

## 2. Materials and Methods

The samples from 49 patients obtained during surgery due to CRC were enrolled in the study. The patients were treated in the 1st Specialistic Hospital in Bytom, Poland (with the approval of the Research Ethics Committee PCN/0022/KB1/42/VI/14/16/18/19/20) between March 2019 and April 2020. The collected specimens included colorectal tumor tissues and surgical margin tissues. Inclusion criteria involved colorectal adenocarcinoma and surgical “tumor-free” margin tissue confirmed by histological examination, patients’ age >18 years, and signed consent. Exclusion criteria included tumors other than adenocarcinoma, presence of margin infiltration, patients’ age <18 years, lack of signed consent, and history of chemo- or radiotherapy. To classify the cancer stage, the TNM staging system and grading were used. The characteristics of the study sample are provided in Table 1.

### 2.1. Preparation of RANTES, PD-L1, IFN-γ, TNF-α, TGF-β, VEGF-A, and VEGF-C Samples for the Evaluation

Fragments of the tumor tissue and surgical margin tissue were weighted and homogenized using a PRO 200 homogenizer (PRO Scientific Inc., Oxford, CT, USA) at 10,000 rpm in nine volumes of phosphate-buffered saline (BIOMED, Lublin, Poland). The suspensions were sonicated with an ultrasonic cell disrupter (UP 100, Hilscher, Ultrasonics GmbH, Teltow, Germany). Subsequently, the homogenates were centrifuged at 12,000 rpm for 5 min at 4 °C. The total protein level was determined using a universal microplate spectrophotometer (μQUANT, Biotek Inc., Winooski, VT, USA).

### 2.2. Evaluation of RANTES, PD-L1, IFN-γ, TNF-α, TGF-β, VEGF-A, and VEGF-C Levels

To assess the levels of the investigated proteins, an enzyme-linked immunosorbent assay (ELISA) was used according to the manufacturer’s instructions. RANTES level was assayed using a human RANTES ELISA kit (Cloud Clone, Wuhan, China) with a sensitivity of 0.059 ng/mL. PD-L1 level was evaluated with a human PD-L1 ELISA kit (Cloud Clone, Wuhan, China) with a sensitivity of 0.057 ng/mL. INF-γ level was determined with a human INFγ level ELISA kit (KHC4022, Invitrogen, Waltham, MA, USA) with a sensitivity of 4 pg/mL. TNF-α level was determined using a human tumor necrosis factor alpha (TNFa) ELISA kit (Cloud Clone, China) with a sensitivity of 5.9 pg/mL. TGF-β level was assessed using a human TGF-β1 ELISA kit (Diaclone, Besancon Cedex, France) with a sensitivity of 8.6 pg/mL. VEGF-A level was determined with a human VEGF-A ELISA kit (Cloud Clone, China) with a sensitivity of 6.2 pg/mL. VEGF-C level was assayed using a human VEGF-C ELISA kit (Biovendor, Brno, Czech Republic) with a sensitivity of 0.057 ng/mL. The absorbance of the samples was determined using a universal microplate spectrophotometer (μQUANT, Biotek Inc., Winooski, VT, USA). The measurement was conducted at a wavelength of 450 nm. The results obtained were recalculated to the corresponding total protein level and presented as ng/mg of protein.

### 2.3. Immunostaining

RANTES and PD-L1 immunostaining was performed in 24 randomly selected cases and CD34 immunostaining was performed in the same 23 cases. The tissue samples were derived from formalin-fixed paraffin-embedded tissue blocks with CRC primary and tumor-free margin specimens. Then the samples underwent deparaffinization and rehydration. Afterward, we performed antigen retrieval by cooking slides in EnVision Flex Target Retrieval Solution High pH (Dako, Carpinteria, CA, USA) for 20 min at 95 °C. The prepared samples were incubated with peroxidase-blocked reagent (Dako, Carpinteria, CA, USA) and then incubated with CD34 antibody (clone: QBEnd/10 Cell Marque, Rocklin, CA, USA; incubation time: 30′; dilution: 1:150), RANTES antibody (clone: A-4, Santa Cruz biotechnology, Heidelberg, Germany; incubation time 60′; dilution 1:200), and PD-L1 (clone ZR3, Cell Marque, Rocklin, CA, USA; incubation time 30′, dilution 1:100). In the next step, the samples were put in EnVision FLEX HRP (Dako, Carpinteria, CA, USA). Then, antigen–antibody complexes were stained using 3,3′-diaminobenzidine. Finally, tissue sections were counterstained with hematoxylin, dehydrated, and covered with coverslips for further analysis.

### 2.4. Histological Evaluation

Histological evaluation was performed by two independent pathologists using an Olympus BX51 microscope.

MVD was assessed on CD34-stained specimens by two independent pathologists using a light microscope in the tumor invasive front regions counting the highest numbers of microvessels per area [15]. Initially, tumor sections were assessed at low magnification to detect tumor invasive front, then three hot spots with high vascularization were chosen. Microvessels were counted in three fields of view under ×20 magnification. MVD was presented as the mean count of microvessels in the assessed view fields; the number was adjusted by the normalization factor (1.210).

TIL assessment was performed in 30 specimens. The percentage of tumor-associated lymphatic infiltration was estimated semi-quantitatively in a four-grade scale on the same H&E-stained slides by the two pathologists, according to the criteria defined by Salgado et al. in breast cancer [16]. These include intratumoral lymphocytes with cell-to-cell contact between lymphocyte and tumor cell and stromal TILs in tumor tissue located dispersed in the stroma within the tumor cells without direct contact, including TILs at the invasive margin. According to the recommendations, stromal TILs were scored as a percentage of the stromal area alone, excluding areas occupied by carcinoma cells. Lymphatic infiltrates outside the tumor borders were not included in the evaluation. Lymphocyte infiltration area lower than 5% was considered TILs 1, whereas 5–25%, 25–50%, and 50–75% of lymphocytes in the stroma were defined as TILs 2, TILs 3 and TILs 4, respectively. More than 75% was defined as TILs 5.

Tumor budding was assessed in the same 30 specimens. Tumor buds were estimated in one FOV at a hotspot area in the invasive front under ×20 magnification. The number of buds was adjusted by the normalization factor (1.210) as described. Budding was reported in the following manner: low budding: 0–4 buds; intermediate budding: 5–9 buds; high budding: >10 buds. The mean number of buds per FOV was also used in the statistical analysis. 

Two independent pathologists assessed RANTES and PD-L1 immunostaining in 24 cases. PD-L1 and RANTES expression was assessed both in tumor cells and in tumor-infiltrating lymphocytes (TILs). PD-L1 and RANTES were considered positive in the tumor cells when staining was present in 1% or more cancer cells. RANTES and PD-L1 expression in TILs was assessed semi-quantitatively in a five-grade scale. An area of lymphocyte infiltration with RANTES or PD-L1 expression lower than 5% was considered as Grade 0, whereas 5–25%, 25–50%, and 50–75% of lymphocytes expressing RANTES or PD-L1 in the stroma were defined as Grade 1, 2, 3 and 4, respectively. More than 75% was defined as Grade 5.

### 2.5. Statistical Analyses

Data distribution was assessed using the Shapiro–Wilk test. The log transformation of the levels of the examined molecules provided a better fit to the Gaussian distribution. Data are presented as mean ± SD for the variables with normal distribution and as median with interquartile range for the variables with non-normal distribution. To compare the tumor and margin levels, the paired Student’s *t*-test (for variables with the normal distribution) and Mann–Whitney U test (for variables with non-normal distribution) were used. Independent variables were also compared using the Student’s *t*-test. To assess the association between RANTES and PD-L1 levels, linear regression was performed. Pearson’s coefficient was used to assess the relationships between the examined variables (including variables with normal distribution). Tau–Kendall’s tau rank correlation coefficient was used for variables with non-normal distribution. Fisher’s exact test was used to test the association between the data obtained from IHC staining for RANTES and PD-L1 expression and clinicopathological parameters of patients. *p* values < 0.05 were considered statistically significant. Statistical analysis was performed using STATISTICA 13 software (Statsoft, Tulsa, OK, USA).) and the ggplot2-R package dedicated to data visualization in RStudio software (Integrated Development for R. RStudio, PBC, Boston, MA, USA). A heatmap with hierarchic clustering was generated using Seaborn (Python data visualization library).

## 3. Results

### 3.1. Results from Tissue Homogenates

We found significantly higher levels of RANTES, PD-L1, IFN-γ, TNF-α, TGF-β, VEGF-A, and VEGF-C in the tumor in comparison with the margin (Table 2 and Table 3, Figure 1 and Figure 2)

The positive association between the concentration of RANTES and PD-L1 was observed in both the tumor and the margin tissue. To assess the relationship between RANTES and PD-L1 levels in the tumor and margin tissue, linear regression analyses were performed (Figure 2).

Furthermore, the tumor levels of RANTES correlated significantly with those of PD-L1, TNF-α, TGF-β, VEGF-A, and VEGF-C. The margin levels of RANTES were significantly associated with the margin levels of all the proteins investigated—PD-L1, IFN-γ, TNF-α, TGF-β, VEGF-A, and VEGF-C (Figure 3, Table 4).

Additionally, we observed that the tumor level of PD-L1 correlated positively with the stage of the disease (Table 5). Furthermore, the tumor level of VEGF-C was positively associated with the value of the T parameter and the stage of the disease (Table 5). Moreover, patients with distant metastases had a significantly higher level of VEGF-C in tumor tissue than those without metastases (Table 6). There were no significant associations between the levels of other investigated proteins and clinicopathological features or between the level of molecules and the patient’s gender or age.

### 3.2. Association with Histologic Features

We obtained HE-stained specimens that allowed us to investigate tumor budding and TILs in 30 randomly selected cases. Additionally, we obtained paraffin-embedded specimens from the tumor to perform CD34, RANTES, and PD-L1 immunostaining. We performed CD34 immunostaining to assess microvessel density in 23 cases; RANTES and PD-L1 immunostaining were performed in 24 cases. The characteristics of MVD in the specimens investigated are presented in Table 7. We did not observe any correlation between MVD, tumor, and margin levels of the molecules investigated and clinicopathological parameters of the patients.

A total of 67% of the investigated cases were characterized by low budding; intermediate or high budding was present in 10 specimens. The mean number of buds was 4.63 +/− 4.45 (range: 0–17 buds) (Table 8). We did not find any association between tumor budding, levels of the investigated proteins, and clinicopathological features of the patients.

TILs were positively associated with IFN-γ level in the tumor and margin and with TGF-β level in the tumor (Table 9). Additionally, TILS negatively correlated with the stage of disease and T feature in the TNM scale (Table 9).

In a randomly selected 24 cases, immunostaining for RANTES and PD-L1 (Figure 4 and Figure 5) was performed. In six different specimens, tumor cells were positive for RANTES and PD-L1. We did not observe any significant associations between RANTES and PD-L1 expression in the tumor, TILs and clinicopathological features of the patients (Table 10, Table 11 and Table 12). 

## 4. Discussion

### 4.1. The Role of RANTES in Tumor Progression

Many studies have reported that inflammation and tumor immunity are crucial for CRC development and progression. Cancer cells are able to create a cytokine network to facilitate immune escape and to support tumor growth. The application of immune checkpoint therapy has been one of the most important events in the anti-cancer fight. The PD-1/PD-L1 immune checkpoint is the main target of this therapy; however, scientists still investigate the potential of other proteins involved in the immune escape because of the toxicity and limitations of the PD-1, PD-L1, and CTLA-4 therapy [17]. For that reason, assessing the cytokine associations becomes an important aim in colorectal cancer studies. Abnormal expression of RANTES has been confirmed in many tumors such as breast, lung, ovarian, prostate, and colorectal cancer [5,6]. Blocking the RANTES pathway was used in the treatment of hematological malignancies and solid tumors: breast cancer, cholangiocarcinoma, gastric, lung, and ovarian cancer [5]. There are also studies assessing the potential of drugs disrupting the RANTES pathway in CRC [5]. It has been demonstrated that the RANTES blockade in CRC results in a decrease in cancer-associated fibroblasts (CAFs) [18], reduced tumor xenograft growth, decreased migration of CRC cells, reduced liver metastases, increased sensitivity to anti-PDGFR therapy, and decreased infiltration of Tregs [19,20]. 

It has been shown that RANTES exhibits pleiotropic effects on the tumor microenvironment through many different mechanisms which affect tumor growth, macrophage polarization, angiogenesis, tumor microenvironment building, migration, and invasion. In the present study we found elevated concentrations of all the investigated proteins in the tumor in comparison to the margin. We also observed significant correlations between the tumor and RANTES margin levels and proteins involved in immune escape and angiogenesis. 

### 4.2. Association between RANTES and PD-L1 Concentrations

Chaio Lu et al. reported that RANTES produced by macrophages is able to stabilize PD-L1 on the surface of CRC cells in vitro and in vivo, thus promoting immune escape in CRC [21]. Furthermore, RANTES induces the formation of nuclear factor kappa-B p65/STAT3 complex which upregulates the promoter of COP9 signalosome 5 (CSN5). CSN5 stabilizes PD-L1 by regulating its deubiquitinating on the cellular level, which results in an increase in PD-L1 activity. In CRC, RANTES not only may directly affect the PD-L1 activity by stabilizing and controlling its deubiquitinating, but is also able to modulate all the components of the tumor microenvironment to promote the immune escape and consequently increase the activity of PD-L1 in the tumor through indirect mechanisms. Furthermore, CCL-5 produced by cancer cells in TME recruits monocytes and educates them to have protumoral, immunosuppressive features. Educated macrophages become M2-TAMS and are also an important source of PD-L1 [5]. In the study presented, we have found a positive correlation between both the tumor and margin levels of RANTES and PD-L1, which seems to confirm the assumptions about the crosstalk between RANTES and PD-L1 pathways.

### 4.3. Correlations between RANTES Levels and Angiogenic Factors: VEGF-A and VEGF-C

In our research, we observed a correlation between the tumor and margin levels of RANTES, VEGF-A, and VEGF-C. It has been confirmed that RANTES may induce the expression of VEGF-A in chondrosarcoma cells [5,22,23]. Furthermore, monocytes and macrophages expressing receptors for RANTES (CCR5) also produce VEGF-A after RANTES stimulation [5]. It has been demonstrated in chondrosarcoma cells that RANTES binds to CCR5 and subsequently activates protein kinase Cδ (PKCδ), c-Src, and hypoxia-inducible factor-1 alpha (HIF-1α), thus inducing VEGF-A-mediated angiogenesis [23]. The involvement of RANTES in tumor angiogenesis is not only limited to promoting VEGF-A production. RANTES has been shown to induce the migration of endothelial cells and to support formation of new vessels. Additionally, RANTES might play some role in lymph vessel development. Li-Hong Wang et al. revealed that RANTES promotes lymphogenesis in human chondrosarcoma cells via stimulation of VEGF-C secretion [23]. A positive association between RANTES and VEGF-C tumor expression was found in chondrosarcoma [23] and is in line with observations in our study. Additionally, we analyzed CD34-stained specimens to assess MVD at the tumor invasive front. We did not observe any correlation between RANTES level and MVD parameters. However, we also did not observe a correlation between VEGF and MVD. One of the reasons for such results may be the small study group (*n* = 23). Moreover, RANTES/CCR5 influence on angiogenesis was mostly studied on chondrosarcoma and human osteosarcoma cells [22,23]. There is no data in the literature regarding the potential role of the RANTES on the formation of the new vessel in colorectal cancer. Despite the fact that our data did not indicate any direct correlations with MVD, the literature provides information indicating the need for in-depth research in this field.

### 4.4. Correlations between RANTES Levels and Cytokines: TGF-β, TNFα, and IFN γ

In the present study we also demonstrated a positive association between RANTES and TGF-β levels in the tumor and margin tissue. Moreover, M. C. Hartmann showed a correlation between the serum concentrations of RANTES and TGF-β in patients with breast cancer [24]. In vitro studies demonstrated that RANTES may stimulate TGF-β production in a direct manner by recruiting macrophages and stimulating them to secrete TGF-β in the tumor microenvironment. It has been reported that melanoma tumor cells exhibit increased expression of the RANTES receptor, CCR5. CCR5 modulates TGF-β activity, which subsequently promotes epithelial to mesenchymal transition and increases the migration of tumor cells [25]. The migration of tumor cells and the influence of RANTES and its receptor CCR5 on this process was investigated in some types of solid cancers as breast, pancreatic, gastric, and colorectal cancer cell lines. Pervaiz et al., in their study on the role of the CCR5/RANTES axis and the effect of its inhibition in colorectal cancer cell lines, confirmed that blocking CCR5 by gene-specific siRNA or low concentrations of maraviroc leads to decreased colony formation and migration of CRC cells [26]. Similar effects were seen in breast cancer cell lines where the effect of RANTES on cell migration under hypoxic conditions was investigated. Hypoxia increased RANTES expression and was responsible for 30% to 50% of hypoxic-mediated cell migration depending on the type of tumor cell line [27]. To assess the influence of RANTES on migration in our study group, we analyzed tumor budding in 23 patients. We did not observe significant correlations. Nevertheless, the data from the literature are very encouraging, and this potential link between RANTES and budding requires more research. In a mouse model of CRC, RANTES stimulated the recruitment of Treg and additionally induced TGF-β expression in Treg cells [19]. We cannot assess which process is responsible for the RANTES/TGF-β correlation observed in our study. Nevertheless, we can be assured that this finding is in line with the data in the literature. 

### 4.5. Correlations between TILS, IFN-γ, and TGF-β Levels

We observed a correlation between TILs and the concentration of IFN-γ both in the tumor and the margin and in the TGF-β tumor concentration. A study conducted by Baker K. et al. indicates that a higher TGF-beta level may predict increased TILs in CRC [28]. 

Many studies have pointed to the significant role of TNFα in colorectal cancer development. High levels of this cytokine were associated with the advanced CRC stage. TNF-α is a central pro-inflammatory cytokine which is secreted with series of inflammatory factors and cytokines by tumor-associated macrophages in the tumor microenvironment [29,30]. Our research results confirm higher levels of TNFα in tumor tissue. Additionally, there was a significant correlation between CCL5 and TNF α levels in the tumor (r = 0.76 *p* < 0.0001) and the margin (r = 0.92, *p* < 0.0001) tissue. 

Previous research reported that TNF-α induced RANTES secretion [31]. Hirano et al. showed that RANTES was induced in human hepatoma cells by treatment with TNF-α via the activation of NF-kappaB and p38 MAP kinase [32]. 

The coordinated expression of CCL5 with TNF-α in our research may point to an inflammatory network between chemokines and cytokines in colorectal cancer and can be associated with the promotion of CCL5 secretion by TNF-α. In our study, we also found a correlation between IFN-γ and CCL5 in the margin tissue (r = 0.5, *p* = 0.002). Liu J et al. proved that IFN may induce RANTES/CCL5 expression in macrophages via a direct, transcriptional activator—Interferon Regulatory Factor 1 (IRF-1). Zhong et al. observed that in cholangiocarcinoma, the presence of TNF-α and IFN-γ stimulated mesenchymal stem cells to secrete CCL5 [5]. 

Researchers have also pointed out that there is a relationship between CCL5 levels and inflammatory molecules. TNF-α and IFN-γ have the potential to enhance CCL5 expression by TLR3 signaling. In this mechanism, TNF-α may activate NF-κB, in cooperation with TLR3 signaling. IFN-γ may stabilize CCL5 mRNA up-regulated by TLR3 [8].

### 4.6. Correlations with Clinical Data

The occurrence of TILs is positively associated with improved overall survival from colorectal cancer [28], probably due to the presence of CD8+ cells and activated T cells in tumor infiltration [19]. Negative correlations between TIL level and clinical data and between stage of disease and N parameter of patients are in line with the results from other studies. High levels of TILs are considered positive prognostic factors in CRC associated with overall survival. High levels of TILs have been confirmed to be related to low TNM stage and histological grade [11,33].

In previous studies, the correlations between the levels of RANTES, PD-L1, IFN-γ, TNF-α, TGF-β, VEGF-A, and VEGF-C in the tumor or serum, and clinicopathological parameters such as TNM classification, histological grade, and poor prognosis were reported, but the results are limited and remain inconclusive. In the present study we found positive associations between VEGF C tumor concentrations, T parameter and clinical stage, as well as between PD-L1 tumor concentration and clinical stage. The positive association between VEGF C expression in the tumor and clinical stage was found in colorectal cancer [34], gastric cancer [35], chondrosarcoma, and lung cancer [36,37]. Similarly, the correlation between PD-L1 expression in the tumor and TNM stage was also described in CRC, but research data are inconclusive [38]. A meta-analysis conducted by Zefeng Shen et al. indicated that the high level of PD-L1 in CRC is positively associated with lymphatic involvement, advanced stage of disease, and poor prognosis [39]. On the contrary, a meta-analysis conducted by Yan Li et al. reported that PD-L1 high expression in CRC is correlated with poor overall survival, shorter DFS, and high-grade tumor, but there was no significant association between PD-L1 level and tumor size, tumor stage, and lymph node involvement [40].

### 4.7. Clinical Implications

Angiogenesis is defined as the formation of new blood vessels from preexisting vessels and has been characterized as an essential process for tumor cell proliferation and viability. The modulation of angiogenesis is highly regulated by proangiogenic and antiangiogenic factors [41,42]. This has led to the development of pharmacological agents for anti-angiogenesis to disrupt the vascular supply and starve tumor of nutrients and oxygen. 

Although we did not prove any direct connections between RANTES and angiogenesis, we obtained data that suggest the need for in-depth study in this area. This necessity of searching for new targets for antiangiogenic therapy is emerging due to the fact of resistance to anti-VEGF therapy. This often occurs because of the escape mechanisms of the angiogenic process through the activation of signaling pathways other than the VEGF pathway [43]. Moreover, it has been proposed that the inhibition of VEGFR by RTKI or an antibody promotes tumor invasiveness and metastasis. Moreover, RANTES might potentially be a good candidate as a therapeutic target in CRC, due to its pleiotropic effect on tumors. However, it might be necessary to evaluate carefully the patient groups that could benefit from RANTES therapy and, as an additional factor, improve anti-PD-L1 therapy by using the effect of PD-L1 stabilization. On the other hand, it might be beneficial for some patients to use anti-RANTES therapy as maraviroc, and to use other compounds discovered in this class which affect the CCR5, the main ligand for RANTES, resulting in, e.g., decreasing cellular proliferation, inducing of apoptosis by arresting cells in the G0/G1 phase, and decreasing cellular migration of the colorectal cancer cells [44]. 

### 4.8. Study Limitations

This study has potential limitations. First of all, it is limited by the small number of patients, the homogenous research group, and its observational character. The data drawn from the study conducted on our group of patients requires further evaluation on cell lines and animal models. 

## 5. Conclusions

RANTES, PD-L1, IFN-γ, TNF-α, TGF-β, VEGF-A, and VEGF-C are upregulated in CRC tissues. RANTES activity might be associated with angiogenesis, lymphogenesis, and immune escape in CRC. Additionally, RANTES might potentially be a good candidate as a therapeutic target in CRC, due to its pleiotropic effect on tumors. 

The future directions of research regarding the role of the RANTES in CRC should focus on its impact on TME modulation, EMT promotion, angiogenesis, and lymphangiogenesis.

## Figures and Tables

**Figure 1 medicina-58-00203-f001:**
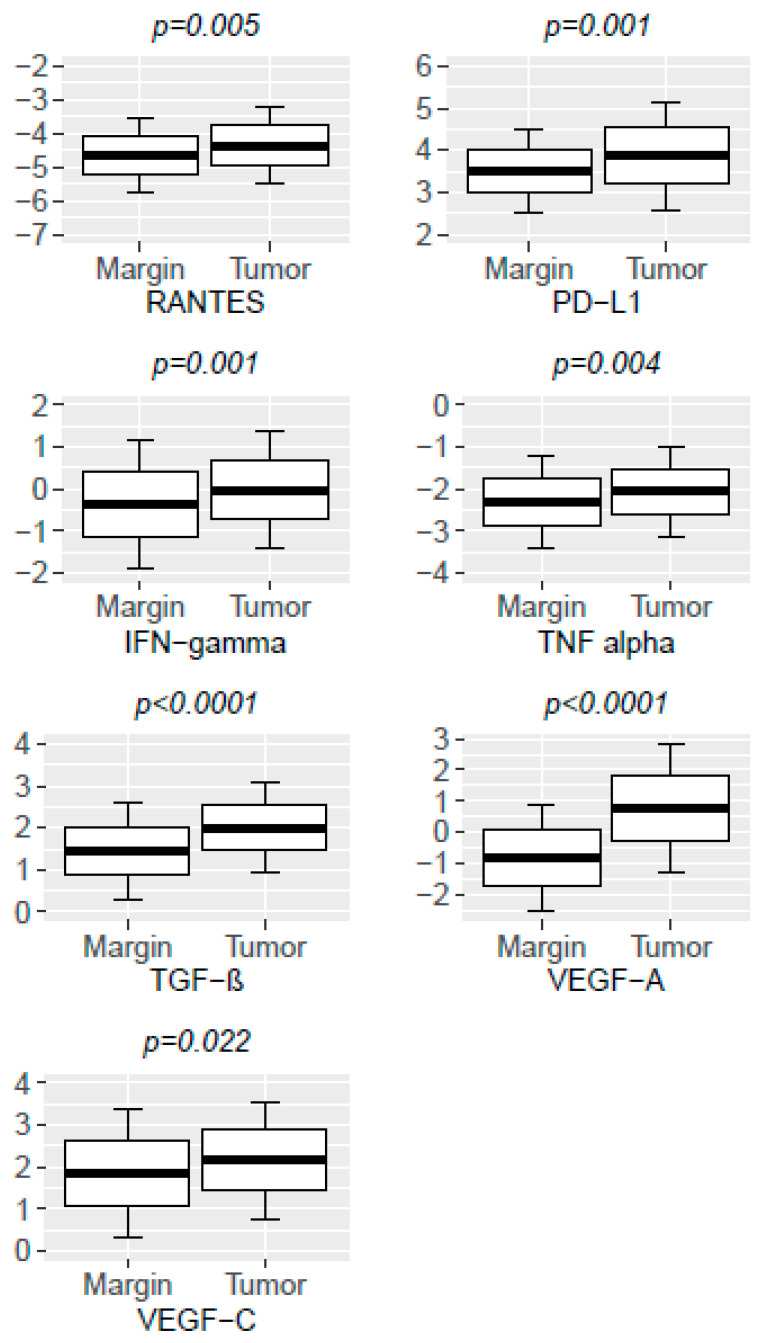
RANTES, PD-L1, IFN-γ, TNF-α, TGF- β, VEGF-A, and VEGF-C levels in the tumor and margin tissues.

**Figure 2 medicina-58-00203-f002:**
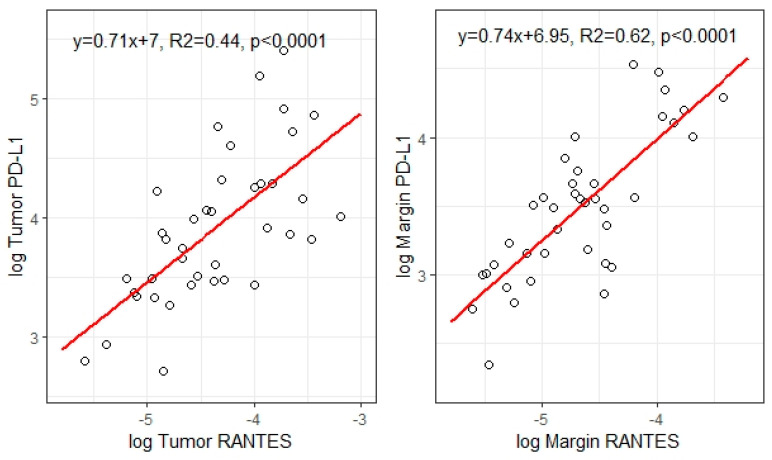
Graphical representation of linear regression results assessing correlations between log RANTES and log PD-L1, respectively, in tumor and margin tissue.

**Figure 3 medicina-58-00203-f003:**
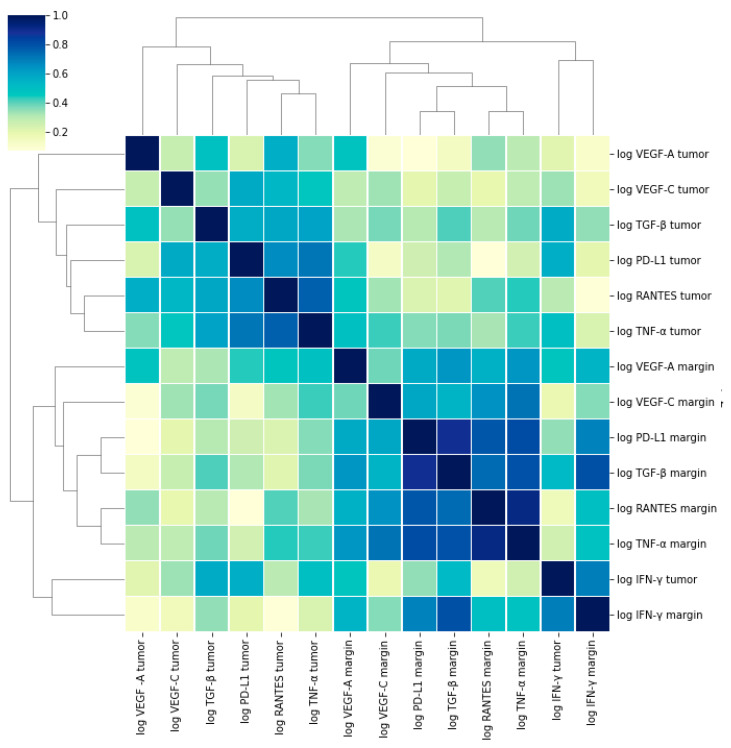
Correlations between the levels of the examined molecules presented as heatmap with hierarchic clustering.

**Figure 4 medicina-58-00203-f004:**
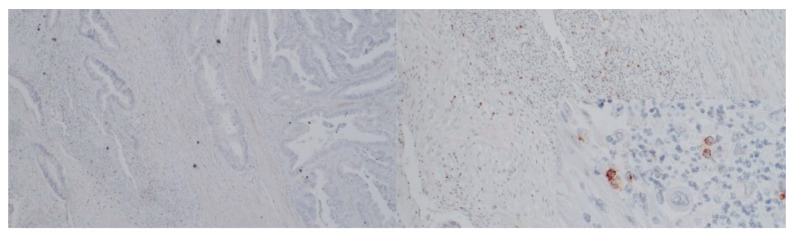
Immunostaining for RANTES (Opta Tech 2200 Camera, magnification 10× and 60×). Cytoplasmatic positive IHC staining for RANTES in tumor-infiltrating lymphocytes.

**Figure 5 medicina-58-00203-f005:**
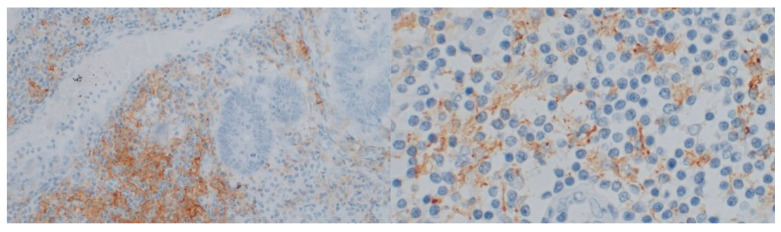
Immunostaining for PD-L1 ((Opta Tech 2200 Camera, magnification 10× and 60×). Cytoplasmatic positive IHC staining for PD-L1 in tumor-infiltrating lymphocytes.

**Table 1 medicina-58-00203-t001:** Characteristics of the patients.

	Female	Male	
	21	28	49 (100%)
Age	62.06 ± 11.31	61.81 ± 9.14	61.92 ± 10.02
T parameter			
T1	0 (0%)	0 (0%)	0 (0%)
T2	7 (33.33%)	5 (17.86%)	12 (24.49%)
T3	11 (52.38%)	14 (50.00%)	25 (51.02%)
T4	3 (14.29%)	9 (32.14%)	12 (24.49%)
N parameter			
N0	9 (42.86%)	12 (42.86%)	21 (42.86%)
N1	9 (42.86%)	9 (32.14%)	18 (36.73%)
N2	3 (14.29%)	7 (25.00%)	10 (20.41%)
M parameter			
M0	18 (85.71%)	19 (67.86%)	37 (75.51%)
M1	3 (14.29%)	9 (32.14%)	12 (24.49%)
TNM stage			
I	6 (28.57%)	4 (14.29%)	10 (20.41%)
II	3 (14.29%)	7 (25.00%)	10 (20.41%)
III	9 (42.86%)	8 (28.57%)	17 (34.69%)
IV	3 (14.29%)	9 (32.14%)	12 (24.49%)
Grading			
G1	1 (4.76%)	0 (0%)	1 (2.04%)
G2	19 (90.48%)	28 (100%)	47 (95.92%)
G3	1 (4.76%)	0 (0%)	1 (2.04%)

**Table 2 medicina-58-00203-t002:** Levels of RANTES, PD-L1, IFN-γ, TNF-α, and TGF-β proteins in tumor and margin presented as log-transformed ng/mg of protein. Paired T-student’s test.

	Tumor		Margin		*p*
	Mean	SD	Mean	SD	
log RANTES	−4.36	0.58	−4.65	0.56	0.005
log PD-L1	3.87	0.66	3.50	0.52	0.001
log IFN-γ	−0.03	0.72	−0.35	0.79	0.001
log TNF-α	−2.06	0.54	−2.31	0.55	0.004
log TGF-β	2.00	0.55	1.45	0.59	<0.0001

**Table 3 medicina-58-00203-t003:** Levels of VEGF-A and VEGF-C proteins in tumor and margin presented as log-transformed ng/mg of protein. Paired T-student’s test.

		Tumor			Margin		*p*
	Median	Q1	Q3	Median	Q1	Q3	
log VEGF-A	0.98	0.21	1.59	−0.62	−1.47	−0.13	<0.0001
log VEGF-C	2.21	1.73	2.68	1.90	1.31	2.41	0.022

**Table 4 medicina-58-00203-t004:** Correlations between the RANTES levels and the examined molecules. R—Pearson’s correlation coefficient.

Pair of Variables	R	*p*
Tumor log RANTES and tumor log PD-L1	0.67	<0.0001
Tumor log RANTES and tumor log IFN-γ	0.30	0.066
Tumor log RANTES and tumor log TNF alpha	0.76	<0.0001
Tumor log RANTES and tumor log TGF-β	0.59	<0.0001
Tumor log RANTES and tumor log VEGF-A	0.57	<0.0001
Tumor log RANTES and margin log VEGF-C	0.54	0.001
Margin log RANTES and margin log PD-L1	0.78	<0.0001
Margin log RANTES and margin log IFN-γ	0.50	0.002
Margin log RANTES and margin log TNF alpha	0.92	<0.0001
Margin log RANTES and margin TGF-β	0.74	<0.0001
Margin log RANTES and margin log VEGF A	0.56	<0.0001
Margin log RANTES and margin log VEGF C	0.65	<0.0001

**Table 5 medicina-58-00203-t005:** Correlations between PD-L1, VEGF-C tumor levels, and clinical parameters of patients (Tau-Kendall’s tau rank correlation coefficient).

Pair of Variables	Tau	*p*
log PDL1 tumor and Stage	0.22	0.04
log VEGF C tumor and T	0.21	0.044

**Table 6 medicina-58-00203-t006:** VEGF-C tumor levels in patients with and without distant metastases. Mann–Whitney U test.

		No Metastases			Metastases		*p*
	Median	Q1	Q3	Median	Q1	Q3	
log VEGF-C	10.97	10.68	20.49	20.56	20.21	20.77	0.0466

**Table 7 medicina-58-00203-t007:** Mean MVD at invasive front in the investigated specimens.

	*n*	Mean	Min	Max	SD
MVD	23	59.57	35.46	89.37	15.80

**Table 8 medicina-58-00203-t008:** Budding and TILs assessment in investigated specimens.

		*n*	%
Budding	0–4	20	66.67
	5–9	5	16.67
	>10	5	16.67
TILs	0–5%	11	36.67
	6–25%	9	30.00
	26–50%	6	20.00
	51–75%	3	10.00
	>75%	1	3.33

**Table 9 medicina-58-00203-t009:** Correlations between TILs, investigated molecules, and clinicopathological parameters of patients.

Pair of Variables	Tau	*p*
TILs and Stage	−0.26	0.047
TILs and N	−0.33	0.01
PD-L1 expression in TILs and TILs	0.53	0.0004

**Table 10 medicina-58-00203-t010:** Clinicopathological parameters of patients according to RANTES expression in the tumor.

Features	Number of Patients	RANTES Tumor-Negative	RANTES Tumor-Positive	*p* Value
	*N* = 24	*N* = 18 (75%)	*N* = 6 (25%)	
Gender				
Male	14 (58%)	10 (56%)	4 (67%)	0.63
Female	10 (42%)	8 (44%)	2 (33%)	
lymph nodes involvement				
yes	14 (58%)	10 (56%)	4 (67%)	0.63
no	10 (42%)	8 (44%)	2 (33%)	
distant metastases				
yes	18 (75%)	4 (22%)	2 (33%)	0.59
no	6 (25%)	14 (78%)	4 (67%)	
pSTAGE				
I/II	10 (42%)	8 (44%)	2 (33%)	0.63
III/IV	14 (58%)	10 (56%)	4 (67%)	
budding				
Grade 1	15 (65%)	11 (61%)	4 (80%)	0.43
Grade 2/3	8 (35%)	7 (39%)	1 (20%)	
TILS				
TILS > 5%	15 (65%)	12 (67%)	3 (60%)	0.78
TILs ≤ 5%	8 (65%)	6 (33%)	2 (40%)	

**Table 11 medicina-58-00203-t011:** Clinicopathological parameters of patients according to PD-L1 expression in the tumor.

Features	Number of Patients	PD-L1 Tumor-Negative	PD-L1 Tumor-Positive	*p* Value
	*N* = 24 (100%)	*N* = 18 (75%)	*N* = 6 (25%)	
Gender				
Male	14 (58%)	12 (67%)	2 (33%)	0.15
Female	10 (42%)	6 (33%)	4 (67%)	
lymph nodes involvement				
yes	14 (58%)	10 (56%)	4 (67%)	0.63
no	10 (42%)	8 (44%)	2 (33%)	
distant metastases				
yes	6 (25%)	5 (28%)	1 (17%)	0.58
no	18 (75%)	13 (72%)	5 (83%)	
pSTAGE				
I/II	10 (42%)	8 (44%)	2 (33%)	0.63
III/IV	14 (58%)	10 (56%)	4 (67%)	
budding				
Grade 1	15 (65%)	10 (59%)	5 (83%)	0.26
Grade 2/3	8 (35%)	7 (41%)	1 (17%)	
TILS				
TILS > 5%	15 (65%)	11 (65%)	4 (67%)	0.93
TILs ≤ 5%	8 (35%)	6 (35%)	2 (33%)	

**Table 12 medicina-58-00203-t012:** Assessment of RANTES and PD-L1 expression in TILs.

		*n*	%
RANTES-positive cells in TILs	0–5%	7	29.17
	6–25%	3	12.50
	26–50%	8	33.33
	51–75%	4	16.67
	>75%	2	8.33
PD-L1-positive cells in TILs	0–5%	9	18.37
	6–25%	7	14.29
	26–50%	5	10.20
	51–75%	1	2.04
	>75%	2	4.08

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
