# Peer review of "Assessment of the RANTES Level Correlation and Selected Inflammatory and Pro-Angiogenic Molecules Evaluation of Their Influence on CRC Clinical Features: A Preliminary Observational Study"

_medicina, 2022, doi:10.3390/medicina58020203_

Round 1

Reviewer 1 Report

Dear Author (s)

  1. Title: "RANTES level and 2 PD‐L1, IFN‐γ, TNF‐α, TGF‐β, VEGF‐A, VEGF‐C and influence 3 of studied proteins...." needs re-edition.
  2. There are several grammatical errors.
  3. Table 1 and its descriptions could be changed to the "result" section.
  4. It is better to reduce the tables and mix some information in a table.
  5.  

Reviewer 2 Report

Major revision:

#1 The heat map in Figure 3 only shows positive correlations.

Were there any negative correlations? Identification of cytokines that are negatively correlated without a significant association may be considered as therapeutic candidates due to their activation.

Minor revisions :

#1        There is no description of the immunostaining.

Line 152: 2.2 Evaluation of RANTES, PD‐L1, IFN‐γ, TNF‐α, TGF‐β, VEGF‐A and VEGF‐C levels

Line 169: 2.3 Immunostaining

The text in the above two sections is the same description.

Please revise the text in Immunostaining.

#2 The resolution is low in figure 2.

It is difficult to read the text in figure 2, please make the text larger.

#3 Line 257

Figure 3 should be changed to Figure 2.

#4  Line 266

Figure 4 should be changed to Figure 3.

#5 Erase the positions of the characters in Line 272 and 273.

#6  There are two different statements regarding the approval number.

Line 131:  PCN/0022/KB1/42/VI/14/16/18/19/20

Line 512:  PCN/0022/KB1/42/VI/14/16/18/

Which is the correct statement?

Round 2

Reviewer 1 Report

Dear Author (S)

There is no problem.

Author Response

Dear Reviewer,

thank you for your comments. We are glad that the evaluation of our manuscript met your expectations and the revised article  has improved in the overall presentation and clarity. 

Sincerely,

Authors

Reviewer 2 Report

This reviewer agrees that the authors have made sufficient revisions.

Author Response

(The authors gave the same response as above.)
